# Influence of Sonication and *Taraxacum Officinale* Addition on the Antioxidant and Anti-ACE Activity of Protein Extracts from *Sous Vide* Beef Marinated with Sour Milk and after In Vitro Digestion

**DOI:** 10.3390/molecules25204692

**Published:** 2020-10-14

**Authors:** Paulina Kęska, Karolina M. Wójciak, Dariusz M. Stasiak

**Affiliations:** Department of Animal Raw Materials Technology, Faculty of Food Science and Biotechnology, University of Life Sciences in Lublin, Skromna 8, 20-704 Lublin, Poland; paulina.keska@up.lublin.pl (P.K.); dariusz.stasiak@up.lublin.pl (D.M.S.)

**Keywords:** beef, bioactive peptides, antioxidant, ACE inhibitor, sonication, dandelion

## Abstract

The present study assessed the effect of pretreating beef as a raw material for sous vide steak preparation. The pretreatment involved maceration of a batch of meat in sour milk with the simultaneous use of ultrasound (250 or 500 W) as well as the addition of *Taraxacum officinale*. The biological activity profile of the peptides was assessed in terms of their antioxidant activity and inhibiting activity against angiotensin-converting enzyme (ACE). Changes in the biological activity of peptides under the influence of hydrolysis by gastrointestinal enzymes, i.e., pepsin and pancreatin, were also considered. There was no significant effect of *T. officinale* addition and sonication of beef batches on the protein content (except for lot S6, after sonication at 500 W as acoustic power and with the addition of dandelion). It was observed that the interaction of maceration in sour milk with simultaneous ultrasound treatment as the initial production step of sous vide beef steak generates the formation of peptides with antioxidant properties. Moreover, peptide formation can be further enhanced by adding dandelion (based on the results of antiradical and chelating activity tests). In addition, the progression of hydrolysis under the influence of gastrointestinal enzymes promotes the release of peptides with antioxidant and anti-ACE activity.

## 1. Introduction

Meat and meat products are considered as main food sources for daily diet. Consumers demand high-quality raw materials, good meat quality with microbial safety, good shelf-life (color and oxidation stability and retention of initial quality), and an appropriate composition (low-fat tissue or intramuscular fat). Additionally, good eating quality (fresh meat appearance and palatability) and ease of preparation is desirable. Meat quality can also be considered in terms of its nutritional density, nutritional value, and nutritional benefit. Several scientific reviews have independently summarized the nutritional value of meat products. Generally, meat as a component of the human diet is a source of nutritional proteins, essential amino acids, vitamin B12, and micro- and macronutrients [1,2]. Meat is also a source of biologically active peptides with pro-health effects on the human body [3,4]. In addition, meat has a low carbohydrate content and does not contain dietary fiber [1,5]. The nutritional, functional, and biological properties of proteins are often influenced by the technology used to process food. It is essential to cook meat to obtain a tasty and safe product; however, the high nutrition value of fresh meat changes under the influence of processing conditions. In particular, oxidation processes for proteins and fats significantly reduce the technical and nutritional quality of meat tissue [6]. Rancidity of fats leads to the formation of secondary oxidation products that pose health risks. For example, these products have been shown to play a significant role as markers of oxidative stress associated with the accelerated aging of body cells [7]. Oxidation has also been shown to cause a number of changes in proteins, which is the main ingredient of meat tissue. Among them, amino acid side chain modification, formation of protein polymers, loss of solubility, increase in carbonyl groups, and changes in amino acid composition are mentioned [8]. These modifications are not only vital for the technical and sensory characteristics of food for muscles, but they can also affect the health and safety of humans after ingestion. For example, it has been observed that cooking increases the production of free radicals, while reducing the antioxidant protection systems in meat that contribute to the oxidation of protein. Sante-Lhoutellier et al. [9] showed a direct and quantitative relationship between protein damage in the food matrix by an oxidative factor (hydroxyl radical) and loss of protein digestibility. It translates into its bioavailability and use as a building component for the body. Interactions between the proteolysis of gastrointestinal enzymes and protein oxidation have been studied, but the results obtained thus far have not been conclusive.

The interest in the use of ultrasound technology in meat processing results from various induced phenomena and effects [10,11]. By selecting the appropriate interaction parameters, these effects are used for diagnostic purposes; process monitoring; and modification of biological, chemical, and physical properties of various media. In the ultrasonic field, substances are subject to accelerated degradation and depolymerization, reactions are catalyzed, and new complex chemical compounds are formed. The course of ultrasound phenomena depends on many factors. They are influenced by the parameters of the ultrasonic field and the initial physicochemical properties of the analyzed materials. Thus, the entirety of the phenomena and effects resulting from the interaction of ultrasonic waves are influenced by the frequency and intensity of ultrasonic waves, the physical structure of the environment and its chemical composition, and, in particular, the efficiency of acoustic coupling [12,13,14].

The causes of such an impact of ultrasound are complex physicochemical processes, which are based on rapidly changing mechanical stresses, energy dissipation, and cavitation together with an entire range of secondary phenomena. In particular, an acceleration of mass transfer processes is observed, and after exceeding the ultrasonic cavitation threshold, cell tissue structures and microorganisms are destroyed [15,16].

Changes caused by cooking meat, in addition to improving the palatability and microbiological quality of products, also lead to quantitative and qualitative loss of meat ingredients, i.e., vitamins, minerals, and water-soluble proteins (so-called cooking loss). The use of the sous vide method, in which raw meat is sealed tightly under vacuum conditions, effectively limits the amount of this meat juice leakage. This process enables the offering of a product with higher nutritional values than that obtained by, e.g., traditional cooking in water [17].

Protection of food ingredients that are important for good health from the harmful effects of oxygen increases the value of food. The use of antioxidant compounds in the production of meat products also applies to compounds of plant origin, partly with the effect of extending the shelf life and health-promoting properties [18]. Among the plant raw materials, dandelion (*Taraxacum officinale*) is a noteworthy plant species. This plant is a perennial from the Asteraceae family. It has been used in phytotherapy, pharmacology, and medicine for a long time because of its health-promoting properties, namely antioxidant, antibacterial, anti-inflammatory, and anticancer properties. Indeed, dandelion contains a wide range of phytochemicals whose biological activity is being actively studied in various areas of human health [19]. The flowers and root are used as a herbal raw material, and the leaves are also used frequently as a raw material in the food industry. Leaf and root extracts show significant antioxidant properties [20]. The antioxidant properties of dandelion are based mainly on the presence of polyphenolic compounds. Ethanol extracts from dandelion leaves contain approximately three times more phenolic compounds (9.9%) and flavonoids (0.086%) than root extracts [21,22]. Other antioxidant compounds present in dandelion are alkaloids, steroids, terpenoids, glycosides, reducing sugars, and tannins. In addition, Biel et al. [20] found that dandelion leaves are a very good source of essential nutrients and elements. They also contain a high amount of phosphorus, potassium, calcium, iron, and zinc, and they are rich in tocopherols, thiamine, riboflavin, and niacin [20], which also indicates the high nutritional value of this plant. It has also been shown that dairy raw materials can be used for meat processing. In particular, fermented dairy products contain a large amount of lactic acid bacteria (LAB). LAB produce compounds that inhibit oxidative processes as well as enhance the production of biologically active peptides from milk proteins through the participation of microbial proteases. Among the natural milk components, proteins and their bioactive peptides offer promising potential for the meat industry. They can affect digestibility (by preventing the oxidation of ingredients in meat products).

In view of the above literature reports, it was decided to include plant raw materials in the production of beef steaks prepared by sous vide methods. It is assumed that the procedure used enhances the antioxidant potential of the compounds present in beef, such as peptides and amino acids. Additional use of technological treatments, namely marinating in sour milk or the use of ultrasound, potentially affects the intensity of biochemical changes (including proteolytic changes of the raw material), thus affecting the degradation of proteins and modifying their biological activity against oxidation (based on ABTS, Fe^2+^ chelation, and reducing power (RP) tests) factors and angiotensin-converting enzyme (ACE)-inhibiting properties.

## 2. Results

### 2.1. Protein-Based Ingredient Changes after Sous Vide Heating and In Vitro Digestion

The influence of bovine meat marinating in sour milk and dandelion herb addition on the content of proteolysis products (such as proteins, peptides, and amino acids) in beef prepared by the sous vide method was determined. The significance levels of the factors included in the experiment and obtained by the two-way ANOVA were defined. Samples (S), proteolytic treatment (T), and the interaction between them (S × T) showed a significant effect (*p* < 0.001). 

The technological treatments used in this study affect the content of proteolysis products as compared to raw beef. A statistically significant (*p* < 0.05) lower protein content was observed in sour milk marinating and heat-treated batches under vacuum packing (S1–S6, average 347.17 µg mL^−1^; Figure 1) compared to nonprocessed raw beef (SC, 1233.08 µg mL^−1^) (*p* < 0.05). This trend was repeated for the peptide content (*p* < 0.05). No significant effect of the type of pretreatment of beef on the protein content was noted, except for batch S6 (after sonication at level II and with the addition of dandelion), which showed a significantly higher protein content than the remaining samples (S1–S5, *p* < 0.05). Furthermore, no differences in peptide content were observed between beef steaks after sous vide heating (*p* > 0.05), except for batch S4 (213.91 µg mL^−1^), which was significantly different (*p* < 0.05) from the nonsonicated batches (S1 and S2; 133.38 and 136.26 µg mL^−1^, respectively) as well as batch S5 (148.37 µg mL^−1^). 

The amino acid content of sous vide beef was also detected. No significant effect (*p* > 0.05) of meat sonication, marinating in sour milk, and maceration with dandelion was observed on the amino acid content, except in a few cases. Among others, the complete absence of cysteine in the batches (S1–S6) was observed. In addition, proline (Pro) and γ-aminobutyric acid (GABA) were present only in samples with the addition of herb (S2, S4, S6; *p* > 0.05). Moreover, an increase in valine (Val), ethanolamine (Eta), and arginine (Arg) in batches with *T. officinale* was observed; however, significant differences were noted only in nonsonicated samples (S1 and S2) (Table 1).

The heating indicates an increase in protein degradation, and the described trends correspond to the results of electrophoretic analysis (Figure 2A). There, the protein extracts were divided into individual fractions (bands) according to the size criterion. The highest number of bands with high color intensity was determined for proteins extracted from the control sample (SC), before marinating and heat treatment. The largest proteins (i.e., 180 kDa band) were also detected only in SC. The most intense bands of approximately 40–50 kDa were most apparent on the SC gel path, and for the remaining batches (S1–S6), the bands were clearly lightened (Figure 2). This observation confirms that thermal processes have a destructive effect on the spatial structure of proteins, thereby causing their breakdown into smaller units (polypeptides, peptides, and amino acids). Differences in the protein profile caused by heat treatment (S1–S6) were also observed in the range of 25–50 kDa, wherein the number of these bands and their intensity were disproportionate. Of the heat-treated batches (S1–S6), the least stained bands were obtained for samples S1 and S2, without additional meat treatment using ultrasound. The electrophoregrams showed no effect of the vegetable additive on the protein content of the final product. As shown in Figure 1, the progressive decrease in the content of proteins was accompanied by an increase in the content of peptides along with the next stage of digestion. SDS-PAGE analysis showed significant protein hydrolysis during digestion with enzymes of the digestive tract (Figure 2B,C). The highest decrease in protein content was observed after one-step hydrolysis with pepsin, which was also illustrated on the electrophoregram (Figure 2B). After digestion with pepsin, bands corresponding to high-molecular-weight proteins disappeared, and only fragments of lower molecular weight were collected at the lower limit of gels; streaks were observed in the 14–18 kDa region (Figure 2B). Samples after two-step hydrolysis (with pepsin and pancreatin) were not detected because analytic material went through the gel in the SDS-PAGE analysis (no particles with mass in the analyzed range were detected) (Figure 2C).

### 2.2. Protein Digestibility during Hydrolysis

A large increase of the protein digestibility in pepsin hydrolysates (from 63.51% for S3 to 78.07% for SC, *p* < 0.05) was observed (Table 2). This enzyme, acting on the structure of proteins, opened subsequent fragments, which made them more available for the operation of the subsequent enzyme, pancreatin. Consequently, the two-stage hydrolysis process caused almost complete degradation of the protein (digestibility of nearly 90%, Table 2), which correlates with the electrophoregram of the analyzed variants (Figure 2B,C). Moreover, among all batches, the raw meat (SC) had the highest digestibility percentage, regardless of the digestion stage. Considering the pretreatment, there were no statistically significant effects of *T. officinale* addition (i.e., in the S2, S4, and S6 batches) or sonication on protein digestibility.

### 2.3. Biological Activity Changes Following In Vitro Digestion

#### 2.3.1. Antioxidant Properties

In the test of the ability of the obtained protein extracts to neutralize free radicals, a decrease was noted in their antiradical activity because of heat treatment. All variants showed lower antiradical activity than the control batches (SC, 72.81%), from 6.6% units for the S4 sample to 33.7% units for the S1 sample (Table 3). A significant effect (*p* < 0.05) of dandelion was observed on the biological activity of protein extracts obtained from the beef steak against ABTS^•+.^ An increase of 12.24% units in activity for variant S2, and an average of 15.95% units for the sonicated samples (S4 and S6) was achieved. This trend was maintained for pepsin and pancreatin hydrolysates (*p* < 0.05). 

The influence of the sonication process on the antiradical activity of the analyzed cooked beef was also confirmed. A significantly (*p* < 0.05) higher activity of protein extracts obtained from batches was noted after ultrasound treatment (cf. S1–S3 and S5). Moreover, the levels of total acoustic power used during sonication did not affect the analyzed parameter, as the results for ultrasound batches were statistically insignificant (Table 3; *p* > 0.05) (cf. S3–S5—sonicated batches without plant additive and S4–S6—sonicated batches with *T. officinale*).

Hydrolytic cleavage of protein chains is an effective method to form biologically active peptides, which were obtained in this study. On the basis of the obtained results, it can be concluded that the progression of hydrolysis under the influence of gastrointestinal enzymes promotes the release of particles with biological activity as measured by the ability to capture ABTS radicals. There was a decrease in the ability of the hydrolysates to neutralize ABTS cations by one-step hydrolysis with pepsin, followed by an increase in activity after two-step digestion (pepsin and pancreatin); this finding is consistent with the trend described in the literature. An increase was observed in the ability of protein extracts to chelate Fe^2+^ activity under the influence of culinary processing of beef steaks (Table 4). After heating at 68 °C under vacuum conditions, all batches showed higher chelation activity (*p* < 0.05) than the control batches (SC). Moreover, a higher difference was observed in batches with the addition of dandelion (S2 and S6; compared to S1 and S5, respectively; *p* < 0.05; Table 4), which indicates the positive effect of the *T. officinale* additive on the analyzed parameter. The influence of ultrasound treatment on the chelating activity of protein extracts was assessed, and the results were found to be ambiguous. Among batches without plant additives, the chelating properties of the protein extract for S3 and S5 batches (after sonication (level I and level II, respectively)) were approximately 10% higher than those for S1 batches (Table 4). A different tendency was noted for the batches after ultrasound treatment and with *T. officinale* addition; compared to S2 batches, the chelating activity of protein extracts was decreased and was approximately 13.5% for S4 batches (lower intensity of sonication) and 5.23% for S6 batches (higher intensity of sonication, Table 4). The effect of in vitro hydrolysis with pepsin and pancreatin was also examined, and the results are shown in Table 4. After pepsin hydrolysis, the ability to chelate Fe^2+^ ions was increased in hydrolysates for nonsonicated batches (S1 and S2) compared to that for batches after ultrasound treatment (S3–S6) (*p* < 0.05). At this stage, there was no significant effect (*p* > 0.05) of the initial meat treatment on the activity of pepsin hydrolysates. The two-stage hydrolysis process (pepsin/pancreatin) led to an increase in the biological activity of the hydrolysis products, with the final value being from 66.61% (S2) to 88.19% (S5). Considering the effect of *T. officinale* addition on the activity of pepsin/pancreatin hydrolysates, a decrease was observed in Fe^2+^ chelating ability (*p* < 0.05) in beef steaks (comparison of S1–S2, S3–S4, and S5–S6; Table 4).

Table 5 shows the results of the reducing power (RP) activity determined for each fraction obtained before and after in vitro hydrolysis. A significant decrease (*p* < 0.05) in RP ability was observed after heat treatment for the S1, S3, and S5 batches (without *T. officinale*) as compared to the control sample (SC, *A_700_* = 0.572). Moreover, the effect of the dandelion additive on RP activity in the extracts and hydrolysates was assessed. All samples showed higher RP activity than the S1 sample. The sonication increased the values of the analyzed parameter in the extracts obtained from beef steaks. This increase was higher in the sample subjected to higher intensities during meat sonication (Table 5). The influence of the sonication process on the RP activity of hydrolysates was ambiguous and depended on the hydrolysis process (single or two step).

#### 2.3.2. Anti-ACE Properties

The protein extracts from control samples (SC) and their hydrolysates showed significantly higher (*p* < 0.05) angiotensin-converting enzyme (ACE-I) activity than the samples treated with preliminary treatments and sous vide cooked, as shown in Table 6. 

There was no influence of the dandelion addition on the analyzed parameter of the protein extracts. Moreover, although the samples sonicated during the pretreatment showed slightly higher values of the ability to inhibit ACE-I, these differences were not significant (*p* > 0.05). On the other hand, an increase in ACE-I-inhibiting activity was observed under the influence of pepsin, and there was a further slight decrease in this activity because of two-stage hydrolysis. A significant (*p* < 0.05) effect of the plant additive was observed in pepsin hydrolysates (cf. S1–S2 and S3–S4) and pepsin/pancreatin (cf. S3–S4 and S5–S6).

## 3. Discussion

Consumption of dietary proteins in proper amounts is important for maintaining good health, and in particular, it is critical to meet the requirements for essential amino acids. Compared to meat, vegetables have a limited amount of protein with sulfur-containing amino acids (Met, Cys, and Trp), which is a disadvantage [23]. The fresh bovine meat used in the present study has high-quality amino acid profiles, namely His (7.37 mg kg^−1^), Thr (8.43 mg kg^−1^), Val (8.99 mg kg^−1^), Lys (17.60 mg kg^−1^), Ile (8.27 mg kg^−1^), Leu (14.73 mg kg^−1^), Phe (7.29mg kg^−1^), Glu (29.6 mg kg^−1^), and Asp (16.97 mg kg^−1^) (unpublished data), which makes it a nutritious food. It has been shown that the use of ultrasound during the initial processing of beef meat did not alter the content of amino acids in the final product. However, an increase in selected essential (His and Val) and nonessential (Pro and Arg) amino acids has been reported in sous vide beef steaks with dandelions. In the analyzed batches, nonprotein amino acids, such as taurine, gamma-aminobutyric acid, and ethanolamine, were also detected, which show bioactive properties according to literature reports [24]. Taurine is a natural component in foods. It is an abundant free amino acid in the cytosol and acts as an antioxidant in various in vitro and in vivo systems with several positive effects on human health [25]. In the present study, the average level of taurine was 0.131mg kg^−1^ and the obtained results were lower than that reported by Purchas et al. [26] (approximately 87.75 mg 100 g^−1^ DM) in cooked beef steaks. According to Purchas et al. [27], the taurine level decreases during beef cooking, and its content depends on the type of muscle [27]. However, in the present study, its level did not depend on the technological treatments used. Similarly, the addition of sour milk (rich in LAB) did not increase the GABA content, although GABA is mainly produced by LAB present in fermented foods [28,29]. GABA has been extensively studied for its numerous physiological functions and positive effects on many metabolic disorders, such as analgesic, anxiolytic, and antihypertensive effects [29]. GABA is ubiquitous among plants [24,30], for example, in mushroom after sous vide cooking [31]. As shown in Table 1, the effect of the plant additive on the GABA content was confirmed; it was present in all samples with dandelion, regardless of the sonication conditions. 

The present study confirmed the influence of thermal treatment and individual in vitro digestion steps on the content of proteins and peptides in marinated beef steaks prepared using the sous vide method, with a high percentage of digestibility of the final product after hydrolysis with pepsin and pancreatin. Among all batches, the raw meat (SC) had the highest digestibility percentage, regardless of the digestion stage. Recent scientific studies indicate the effect of heat treatment on the formation of protein aggregates. Their effect was shown on the digestibility of proteins from processed products, such as meat, which confirmed their lower digestibility in vivo and/or in vitro. For example, Li et al. [32] studied differences in the protein digestibility of four pork products. They indicated technological treatments as the main source of variability in meat digestibility. As an example, long-term braising promotes protein oxidation and aggregation. The meat cooking process can produce chemical reactions that affect several amino acids (carbonyl formation, thiol oxidation, and aromatic hydroxylation), leading to protein denaturation, protein cross-linking, and protein aggregate formation [32]. Sarcoplasmic proteins in particular, which include different types of enzymes, pigments, and regulatory proteins, can aggregate between 40 and 60 °C. Heat-induced oxidation and increased surface hydrophobicity of myofibrillary proteins may also cause protein aggregation, which may affect their proteolytic susceptibility (i.e., digestibility). At 70 °C (similar to the condition used in the present study), moderate denaturation of meat proteins occurs [32], although Bax et al. [33] indicated that high temperatures (above 100 °C) improve overall meat protein digestibility. In addition, in the present study, S1 and S2 (nonsonicated samples) yielded the lowest values of intensities for bands in the electrophoregram and showed results different from sonicated samples (S3–S6). This finding indicates a different profile of protein-derived products in the extracts (increased proportion of smaller protein degradation products) in these samples. This result shows the effect of ultrasound on the structure of proteins. The applied pretreatment of meat by sonication may probably cause changes in the structure of the protein chain, resulting in lower access for proteins in beef steak. This observation is different from that reported in some previous studies. Vidal et al. [34] observed no effect of ultrasound (use of prior or concomitant ultrasound treatment) on the degree of hydrolysis of porcine and bovine collagen under the influence of pepsin (4% enzyme addition). On the other hand, it was shown that the simultaneous action of ultrasound and 8% enzyme concentration (Alcalase 2.4 L) proved to be the most effective in the bovine collagen sample hydrolysis process, providing the best degree of hydrolysis and antioxidant activity results [35]. However, Leong et al. [36] reported that ultrasound treatment breaks down aggregates of casein and whey proteins, resulting in an increased availability of these proteins for proteolytic enzymes. Munir et al. [37] also reported that an initial sonication treatment increases the proteolysis rate during the production of cheddar cheese and its subsequent maturation. Regarding the pretreatment, there was no significant effect of *T. officinale* addition (in the S2, S4, and S6 batches) or sonication condition (acoustic power level) on protein digestibility. As reported by Dong et al. [38], in vitro digestibility of shrimp proteins is dependent on the processing time. The authors indicated sonication for 20 min (other parameters were 20 kHz frequency and 400 W power) was effective in the modification of the in vitro digestibility of shrimp proteins. Thus, the lack of changes in the digestibility of proteins from beef steaks in the sonicated samples can probably be explained by the very short time of the ultrasound treatment used in the present study.

Contemporary knowledge in the field of meat technology indicates a number of possible directions of modification of the nutritional and health-promoting value of meat products. This can be achieved, for example, through innovative technological treatments (e.g., sonication), adding an ingredient with high pro-health potential (e.g., dandelion herb), or discovering new properties of food ingredients, such as biologically active peptides. In this context, these peptides can be used as new ingredients in the development of functional food products. Food-derived bioactive peptides contain a sequence of 2–20 amino acids and can exhibit various in vivo effects [39,40]. Among the many types of biopeptide activity, their antioxidant and angiotensin-converting enzyme (ACE) inhibitory properties are the most frequently studied due to their potential in preventing chronic non-communicable diseases [3,4]. Peptides acting as antioxidants may contribute to the maintenance of the oxidative stability of the meat tissue. They are also involved in preventing the negative effects of oxidative stress. This is due to the excessive accumulation of reactive oxygen species in the body and can lead to pathological conditions manifested by cardiovascular diseases, diabetes, and other metabolic disorders. In turn, peptides with ACE inhibitor activity contribute to the reduction of blood pressure in patients with arterial hypertension, which is a risk factor for the development of cardiovascular diseases. The study assessed the effect of peptides obtained from sous vide beef steaks in relation to the above-mentioned bioactivities. It was found that the proteolytic changes caused by the applied technological measures favored the release of peptides acting as antioxidants and inhibiting the action of ACE. The literature reports the use of the ultrasound treatment process in modifying the biological activity of peptides from various food sources. Gao et al. [41] assessed the effect of sonication on the antioxidant activity of compounds in soy sauce during moromi fermentation. The authors showed that sonication increased the antioxidant properties of soy sauce, which was attributed, among others, to a significantly higher level of free amino acids and a large number of small peptides in the sonicated moromi. Another study indicated that free amino acids contributed to increased metal ion chelating activity and DPPH radical scavenging activity [37]. However, in the present study, pretreatment of meat with ultrasound during marinating with sour milk had a low effect on the variation in the content of free amino acids in trials. On the other hand, the sonication process contributed to the increased number of peptides and increased antioxidant activity as determined by the ABTS, RP, and chelation tests (in the trials without the addition of the dandelion herb). According to Uluko et al. [42], the shear forces generated by acoustic cavitation during ultrasonic treatment can hydrolyze proteins, leading to the release of bioactive peptides. Nevertheless, as shown earlier [43], treatment with ultrasound as a preliminary stage in the preparation of raw pork meat tissue (as a process preceding maceration of loins in acid whey) did not promote the release of antioxidant or ACE-I inhibitory peptides. This relationship suggests the use of simultaneous maceration in sour milk and the action of ultrasound lead to the formation of biologically active peptides in meat tissue. Nevertheless, in order to produce a specific physiological effect, potentially bioactive compounds delivered to the human body with food must be resistant to degradation by digestive enzymes in the gastrointestinal tract and reach the appropriate target in an active form. Therefore, the bioavailability and biological activity of the peptides were also assessed under simulated gastrointestinal conditions. The influence of the sonication process on biological activity was also noted in pepsin and pancreatin hydrolysates after in vitro digestion, but the result was inconclusive and depended on the degree of hydrolysis (one or two step).

## 4. Materials and Methods 

### 4.1. Preparation of Beef Steak

*Musculus semimembranosus* obtained from Limousin heifer with a body weight of approximately 400–450 kg and age of 30 months was excised at 24 h postmortem. The meat that originated from breeding was certified as organic by the Polish certifying body according to the Commission Regulation (EC) No. 889/2008 of 5 September 2008 that established detailed rules for the implementation of Council Regulation (EC) No. 834/2007 on organic production and labeling of organic products with regard to organic production, labeling, and control (http://data.europa.eu/eli/reg/2008/889/2020-01-07). The meat used for the research was prepared by cleaning the surface of the fascia, remaining tendons, etc. The protruding fragments of the muscle were removed, giving the meat portion a uniform oval shape; this was then washed with a stream of tap water. In the next stage, each portion from seven muscles was cut into eight slices of 3 cm thickness and weighing 150 ± 10 g, perpendicular to the direction of the muscle fibers. The slices were stored in plastic bags for 48 h at 4 °C. Six groups of muscles were marinated in organic sour milk (PL-EKO-09) for 24 h at 4 °C (S1–S6). Four muscle samples (S3–S6) were additionally treated in an ultrasonic laboratory batch processor (Polsonic, Warsaw, Poland). The wave frequency was 40 kHz. Two levels of total acoustic power were applied: level I was approximately 250 W (S3–S4) and level II was approximately 500 W (S5–S6). The muscle samples immersed in the chilled sour milk were treated for 4 min. The milk temperature was increased on average by 14 ± 1 °C due to dissipation. Therefore, sour chilled milk (5 ± 1 °C) was used each time before starting and during the treatment. The slices were then placed in separate plastic bags (cooking bags 80GR vacuum cooking, Orved S.P.A, Venezia, Italy). The common dandelion (*T. officinale*) herb (Polskie Zioła, Piaski Wielkie, Poland) was added to the samples S2, S4, and S6 in a 1% (*w*/*w*) proportion. The control group consisted of two pieces of meat: one without any addition (SC) and the other marinated in sour milk (S1) (Table 7). All bags were then vacuum-sealed (VAC-20 DT, Edesa Hostelera S. a., Barcelona, Spain) and were sous vide heat treated in water in a PolyScience heated circulating bath with a digital controller at 63 ± 0.5 °C for 90 min.

After sous vide treatment, the samples were cooled in water at 2 °C for 1 h and stored at 4 °C for 24 h. The analysis was performed using three independent series of tests, with three replicates each (*n* = 9).

### 4.2. Amino Acid Content 

The content of free amino acids (in sous vide beef steak; S1–S6) was determined using the automatic amino acid analyzer AAA 400 (Ingos Ltd., Praha 4-Komořany, Czech Republic) equipped with the Ostion LG ANB ion exchange column (36 × 0.37 cm) kept at 70 °C according to the method described by Stadnik and Dolatowski [44]. The following amino acids in sous vide beef were considered: taurine (Tau), aspartic acid, threonine, serine, glutamic acid, proline, glycine, alanine, valine, cysteine, methionine, isoleucine, leucine, tyrosine, phenylalanine, ethanolamine, ornithine, γ-aminobutyric acid, lysine, histidine, 1-methylhistidine, and arginine and were expressed as mg kg^−1^.

### 4.3. In Vitro Hydrolysis of Protein Extracts

Water-soluble protein extracts were obtained according to Molina and Toldrá [45]. Samples (10 g) were extracted with 100 mL of phosphate buffer solution (15.6 mM Na_2_HPO_4_, 3.5 mM KH_2_PO_4_, pH 7.5) using a homogenizer (IKA T25, Staufen, Germany) at 8000 rpm for 1 min in an ice bath. The homogenate was centrifuged at 5000× *g* for 20 min at 4 °C. Next, the pH of the extracts was adjusted using 1M HCl (to pH 2.0–2.5) and pepsin (conditions: enzyme: protein solution 1:100 (*w*/*w*), 2 h at 37 °C and with constant stirring) was added. The solution was neutralized to pH 7.0 with 1 M NaOH, and pancreatin was added at the enzyme to protein ratio of 1:50 [45]. After each hydrolysis step, the samples were heated (100 °C for 10 min). To remove nonhydrolyzed proteins, all digesta were centrifuged at 10,000× *g* at 4 °C for 20 min. The supernatant was filtered, and the peptide content was determined according to the method described by Adler-Nissen [46] using 2,4,6-trinitrobenzenesulfonic acid (TNBS). The results were determined on the basis of a standard curve prepared for L-leucine as the reference amino acid. The supernatant was stored at −60 °C for further determination of antioxidant and ACE-inhibiting activities. The content of proteins in the precipitate was determined by Bradford’s method [47]. The results were calculated using a standard curve prepared for bovine serum albumin (BSA) as a reference protein. 

The degree of digestibility was determined by the formula:Digestibility [%] = [PC_0_ − PC_1_)]/PC_0_(1)
where PC_1_—content of proteins in the precipitate after digestion with the enzyme of the gastrointestinal tract; PC_0_—protein content in nondigested samples.

### 4.4. Gel Electrophoresis 

The nondigested samples as well as pepsin and pancreatin hydrolysates were separated on SDS-PAGE gels according to the Laemmli method [48]. The electrophoretic separation conditions were as follows: a 5% surface gel and a 14% separation gel at a constant current of 50 V for the surface gel and 100 V for a release gel. The analysis was performed using the Mini Protean Tetra Cell (Bio-Rad, Hercules, CA, USA). A mass marker in the 14–198 kDa range (Invitrogen, Carlsbad, CA, USA) was used in the comparative analysis of bands.

### 4.5. Antioxidant Properties

#### 4.5.1. Antiradical Activity

The ability of the obtained extracts or hydrolysates to eliminate free radicals was determined by the method of Re et al. [49] using the free radicals of ABTS. The presence of antioxidant in the solutions results in the reduction of ABTS•* to ABTS and subsequent discoloration, the degree of which is proportional to the content of antioxidant. The ABTS•* reduction rate was determined spectrophotometrically (U-5100 UV-Vis, Hitachi HighTechnologies America Inc., Schaumburg, IL, USA) at 734 nm. The ability to neutralize free radicals (antiradical) was determined by the formula:Scavenging [%] = [1 − (As/Ac)] × 100(2)
where As—absorbance of sample; Ac—absorbance of control (ABTS solution).

#### 4.5.2. The ability to Chelate Iron Ions (II)

The test of iron (II) ions’ chelating ability of the compounds contained in the extracts was performed according to the Decker and Welch [50] method. The absorbance of the colored complex was measured spectrophotometrically at the wavelength of 562 nm by using the formula:Chelation activity [%] = [1 − (As/Ac)] × 100(3)
where As—absorbance of sample; Ac—absorbance of control.

#### 4.5.3. The Power to Reduce Iron Ions (III)

RP was determined using the method of Oyaizu [51] in which the reduction of the reagent (Fe^3+^) in a stoichiometric excess is based on the amount of antioxidants. Antioxidant compounds cause reduction of the iron form (Fe^3+^) to the ferrous form (Fe^2+^) because of their reduction ability, which manifests as a blue-colored complex. The reduction can thus be determined spectrophotometrically at 700 nm. The higher the absorbance value, the greater is the ability to reduce the test substance.

### 4.6. ACE-Inhibiting Properties

Angiotensin-converting enzyme-I (ACE-I)-inhibiting activity was measured by the spectrophotometric method according to Nasri et al. [52]. The absorbance was measured at 390 nm, and the ACE-I inhibiting activity was calculated using the following equation: ACE inhibition [%] = [(B − A)/(B − C)] × 100,(4)
where A is the absorbance of extract generated in the presence of ACE inhibitor component, B is the absorbance of extract generated without ACE inhibitors, and C is the absorbance of extract generated without ACE.

### 4.7. Statistical analysis

All results are expressed as mean ± standard deviation (SD). The differences between groups were assessed using one-way analysis of variance and Tukey’s multiple-range test. The analysis was preceded by a test of homogeneity and normality of variance. Mean values were considered significantly different at the *p* value of <0.05, e.g., *** (*p* < 0.001), ** (*p* < 0.01), * (*p* < 0.05), using Statistica^®^ 13.1 software (StatSoft Polska Sp. z o. o., Cracow, Poland).

## 5. Conclusions

The results of the present study showed that there is a possibility to produce potentially functional meat products as a result of the use of ultrasound in combination with dandelion extracts. The potentially functional characteristics of meat product probably derive from the presence of biologically active peptides increased by sonication and polyphenols from *T. officinale*.

The use of a *T. officinale* additive during sous vide processing enhances the antioxidant potential of compounds present in beef, such as peptides and amino acids. Additionally, the use of dandelion increases the potential of beef steaks prepared with the sous vide method against oxidative factors. The use of ultrasound affects the intensity of proteolytic changes of the product. It was observed that the interaction of ultrasound treatment during the maceration of meat with sour milk may contribute to an increase in the antioxidant potential and inhibition of ACE. Importantly, the obtained protein extracts also showed biological activity under the influence of proteolytic enzymes, which are the equivalent of human digestive enzymes. Therefore, the results of the present study may have future applications for the production of healthier meat-based products.

## Figures and Tables

**Figure 1 molecules-25-04692-f001:**
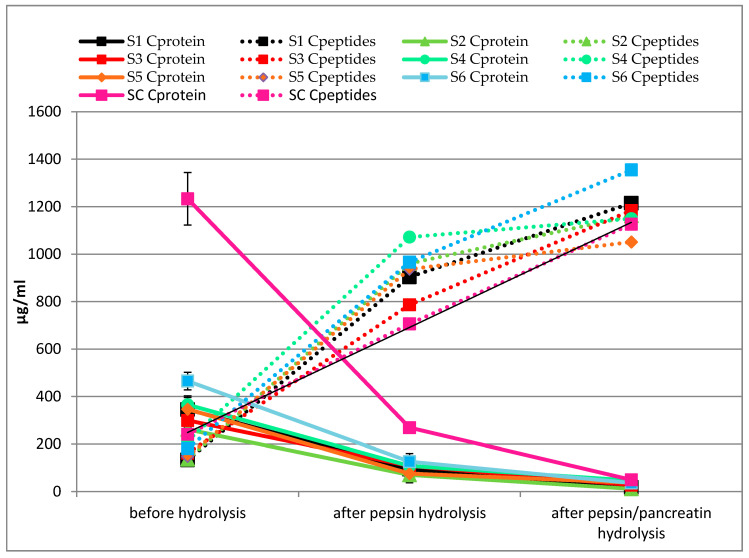
Protein and peptide content (µg mL^−1^) during in vitro hydrolysis of cooked beef steaks.

**Figure 2 molecules-25-04692-f002:**
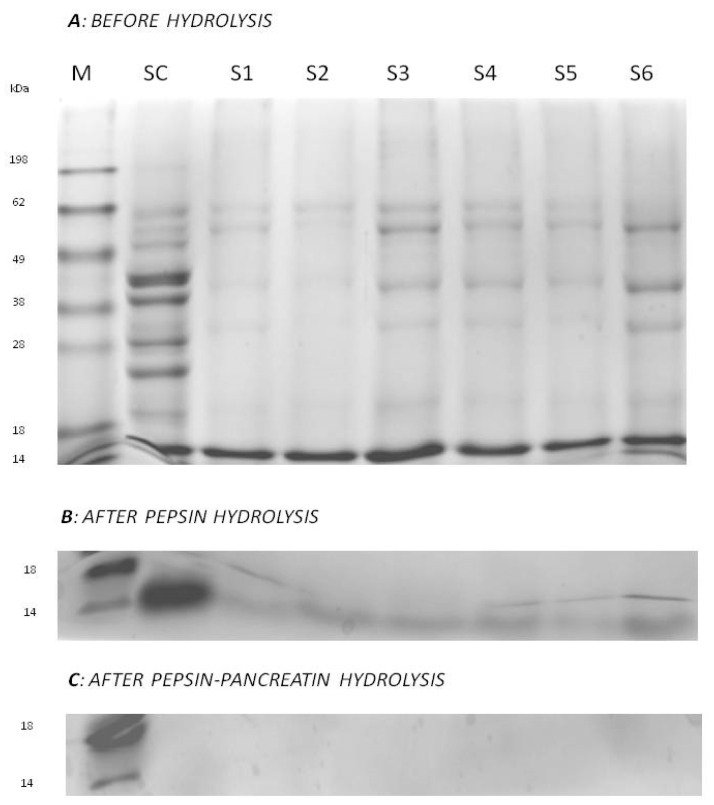
The SDS-PAGE profile of proteins from beef steak obtained before hydrolysis (**A**), after pepsin hydrolysis (**B**) and after pepsin-pancreatin in vitro hydrolysis (**C**). M—molecular weight marker; SC—control samples; S1–S6—analyzed samples.

**Table 1 molecules-25-04692-t001:** Amino acid content in sous vide beef steak (mg kg^−1^).

Amino Acid	S1	S2	S3	S4	S5	S6
*Essential amino acids (EAAs)*						
His	0.019 ± 0.003 ^a^	0.047 ± 0.012 ^b^	0.027 ± 0.003 ^ab^	0.041 ± 0.014 ^ab^	0.024 ± 0.004 ^ab^	0.034 ± 0.009 ^ab^
Thr	0.016 ± 0.010 ^a^	0.039 ± 0.012 ^a^	0.021 ± 0.010 ^a^	0.032 ± 0.010 ^a^	0.028 ± 0.004 ^a^	0.031 ± 0.008 ^a^
Val	0.037 ± 0.009 ^a^	0.078 ± 0.016 ^b^	0.053 ± 0.004 ^ab^	0.076 ± 0.021 ^ab^	0.056 ± 0.011 ^ab^	0.071 ± 0.019 ^ab^
Met	0.024 ± 0.005 ^a^	0.031 ± 0.010 ^a^	0.024 ± 0.005 ^a^	0.032 ± 0.013 ^a^	0.029 ± 0.003 ^a^	0.030 ± 0.012 ^a^
Lys	0.039 ± 0.005 ^a^	0.091 ± 0.032 ^a^	0.053 ± 0.004 ^a^	0.082 ± 0.033 ^a^	0.047 ± 0.004 ^a^	0.071 ± 0.031 ^a^
Ile	0.028 ± 0.008 ^a^	0.051 ± 0.012 ^a^	0.034 ± 0.005 ^a^	0.047 ± 0.018 ^a^	0.037 ± 0.006 ^a^	0.049 ± 0.01 ^a^
Leu	0.060 ± 0.011 ^a^	0.105 ± 0.025 ^a^	0.079 ± 0.004 ^a^	0.104 ± 0.037 ^a^	0.086 ± 0.010 ^a^	0.109 ± 0.029 ^a^
Phe	0.035 ± 0.015 ^a^	0.064 ± 0.014 ^a^	0.044 ± 0.003 ^a^	0.066 ± 0.022 ^a^	0.054 ± 0.009 ^a^	0.061 ± 0.010 ^a^
*Nonessential amino acids (NEAAs)*						
Asp	0.002 ± 0.003 ^a^	0.010 ± 0.005 ^a^	0.004 ± 0.003 ^a^	0.005 ± 0.004 ^a^	0.009 ± 0.010 ^a^	0.017 ± 0.008 ^a^
Ser	0.027 ± 0.009 ^a^	0.055 ± 0.011 ^a^	0.030 ± 0.013 ^a^	0.053 ± 0.023 ^a^	0.046 ± 0.005 ^a^	0.055 ± 0.019 ^a^
Glu	0.305 ± 0.097 ^a^	0.527 ± 0.050 ^a^	0.431 ± 0.054 ^a^	0.488 ± 0.182 ^a^	0.401 ± 0.033 ^a^	0.380 ± 0.121 ^a^
Gly	0.035 ± 0.015 ^a^	0.049 ± 0.006 ^a^	0.039 ± 0.001 ^a^	0.047 ± 0.016 ^a^	0.048 ± 0.005 ^a^	0.042 ± 0.010 ^a^
Pro	n.d.	0.136 ± 0.036 ^a^	n.d.	0.142 ± 0.076 ^a^	n.d.	0.110 ± 0.047 ^a^
Ala	0.139 ± 0.044 ^a^	0.213 ± 0.021 ^a^	0.173 ± 0.011 ^a^	0.214 ± 0.072 ^a^	0.189 ± 0.016 ^a^	0.174 ± 0.042 ^a^
Arg	0.028 ± 0.011 ^a^	0.068 ± 0.019 ^b^	0.023 ± 0.002 ^a^	0.030 ± 0.013 ^a^	0.034 ± 0.011 ^a^	0.051 ± 0.007 ^ab^
Cys	n.d.	0.009 ± 0.016	n.d.	n.d.	n.d.	n.d.
Tyr	0.030 ± 0.014 ^a^	0.051 ± 0.024 ^a^	0.037 ± 0.011 ^a^	0.053 ± 0.017 ^a^	0.058 ± 0.015 ^a^	0.046 ± 0.025 ^a^
EAA/NEAA ratio	0.466 ± 0.061 ^a^	0.456 ± 0.089 ^a^	0.457 ± 0.055 ^a^	0.478 ± 0.124 ^a^	0.460 ± 0.054 ^a^	0.524 ± 0.025 ^a^
*Nonprotein amino acids*						
1mHis	0.257 ± 0.074 ^a^	0.246 ± 0.036 ^a^	0.273 ± 0.065 ^a^	0.297 ± 0.061 ^a^	0.278 ± 0.053 ^a^	0.201 ± 0.0238 ^a^
GABA	n.d.	0.021 ± 0.004 ^a^	n.d.	0.013 ± 0.004 ^a^	n.d.	0.015 ± 0.004 ^a^
Eta	0.000 ± 0.000 ^a^	0.023 ± 0.003 ^bc^	0.012 ± 0.003 ^b^	0.018 ± 0.005 ^bc^	0.016 ± 0.002 ^bc^	0.027 ± 0.007 ^c^
Orn	0.003 ± 0.005 ^a^	0.006 ± 0.011 ^a^	0.020 ± 0.003 ^a^	0.028 ± 0.009 ^a^	0.012 ± 0.012^a^	0.008 ± 0.011 ^a^
Tau	0.090 ± 0.007 ^a^	0.156 ± 0.044 ^a^	0.131 ± 0.045 ^a^	0.142 ± 0.046 ^a^	0.154 ± 0.043 ^a^	0.114 ± 0.021 ^a^

^a–c^ Within the same sample (column), mean values followed by the common letter do not differ significantly (*p* < 0.05). n.d.—not detected. S1—sample marinated in sour milk; S2—sample marinated in sour milk and with addition of herbs; S3—sample after ultrasonoud (250 W) treatment and marinated in sour milk; S4—sample after ultrasonoud (250 W) treatment and marinated in sour milk and with addition of herbs; S5—sample after ultrasonoud (500 W) treatment and marinated in sour milk; S6—sample after ultrasonoud (500 W) treatment and marinated in sour milk and with addition of herbs. His—histidine, Thr—threonine; Val—valine; Met—methionine; Lys—lysine; Ile—isoleucine; Leu—leucine; Phe—phenylalanine; Asp—aspartic acid; Ser—serine; Glu—glutamic acid; Gly—glycine; Pro—proline; (Gly), Ala—alanine; Arg—arginine; Cys—cysteine; Tyr—tyrosine; 1mHis—1-methylhistidine; GABA—γ-aminobutyric acid; Eta—ethanolamine; Orn—ornithine, Tau—taurine.

**Table 2 molecules-25-04692-t002:** Protein digestibility (%) from sous vide beef steak after hydrolysis with gastrointestinal enzymes.

Sample	Pepsin Hydrolysates	Pepsin/Pancreatin Hydrolysates	Sample (S)	Treatment (T)	S × T
S1	72.09 ± 5.17 ^Aab^	94.63 ± 2.12 ^Bac^	***	***	*
S2	73.13 ± 4.30 ^Aa^	85.61 ± 5.51 ^Bb^			
S3	63.51 ± 3.41 ^Ab^	90.37 ± 5.16 ^Babc^			
S4	72.84 ± 9.63 ^Aab^	86.97 ± 4.79 ^Bb^			
S5	71.65 ± 4.56 ^Aab^	89.07 ± 3.95 ^Bab^			
S6	72.84 ± 4.12 ^Aa^	92.18 ± 1.70 ^Bab^			
SC	78.07 ± 2.40 ^Aa^	96.03 ± 1.18 ^Bc^			

^a–c^ Within the same sample (column), mean values followed by the common letter do not differ significantly (*p* < 0.05). ^A,B^ Within the proteolytic treatment (row), mean values followed by the common letter do not differ significantly (*p* < 0.05); mean ± standard deviation; *** (*p* < 0.001), * (*p* < 0.05).

**Table 3 molecules-25-04692-t003:** Results of radical scavenging activity against ABTS^•+^ (%).

Sample	Nondigested Extract	Pepsin Hydrolysate	Pepsin/Pancreatin Hydrolysate	Sample (S)	Treatment (T)	S × T
S1	39.11 ± 5.54 ^Aa^	33.33 ± 3.22 ^Aa^	87.33 ± 2.17 ^Ba^	***	***	***
S2	51.35 ± 4.15 ^Ab^	44.44 ± 2.68 ^Bb^	98.72 ± 0.38 ^Cb^			
S3	49.37 ± 5.12 ^Ab^	13.40 ± 2.58 ^Bc^	85.36 ± 0.70 ^Ca^			
S4	66.21 ± 5.14 ^Ac^	41.73 ± 6.82 ^Bb^	97.99 ± 1.01 ^Cb^			
S5	49.47 ± 4.36 ^Ab^	30.32 ± 0.98 ^Ba^	79.68 ± 1.39 ^Cc^			
S6	64.52 ± 3.26 ^Ac^	41.63 ± 4.59 ^Bb^	96.63 ± 2.99 ^Cb^			
SC	72.81 ± 2.00 ^Ac^	55.60 ± 2.53 ^Bd^	99.01 ± 0.71 ^Cb^			

^a–d^ Within the same batches (column), mean values followed by the common letter do not differ significantly (*p* < 0.05). ^A–C^ Within the proteolytic treatment (row), mean values followed by the common letter do not differ significantly (*p* < 0.05). Mean ± standard deviation. *** (*p* < 0.001).

**Table 4 molecules-25-04692-t004:** Results of Fe^2+^chelating activity (%).

Sample	Nondigested Extract	Pepsin Hydrolysate	Pepsin/Pancreatin Hydrolysate	Sample (S)	Treatment (T)	S × T
S1	17.97 ± 0.165 ^Aa^	33.45 ± 1.35 ^Ba^	82.65 ± 0.24 ^Ca^	***	***	***
S2	38.31 ± 3.22 ^Ab^	31.77 ± 2.24 ^Ba^	66.61 ± 0.65 ^Cb^			
S3	26.37 ± 2.02 ^Acd^	17.85 ± 1.17 ^Bb^	83.50 ± 0.83 ^Ca^			
S4	24.82 ± 2.85 ^Ad^	11.57 ± 3.55 ^Bb^	77.48 ± 0.25 ^Cc^			
S5	27.72 ± 2.48 ^Ad^	13.61 ± 3.39 ^Bb^	88.19 ± 0.97 ^Cd^			
S6	33.08 ± 0.43 ^Abc^	17.75 ± 3.20 ^Bb^	84.26 ± 1.13 ^Ca^			
SC	10.34 ± 2.99 ^Ae^	17.43 ± 0.96 ^Bb^	75.61 ± 1.24 ^Cc^			

^a–e^ Within the same batches (column), mean values followed by the common letter do not differ significantly (*p* < 0.05). ^A–C^ Within the proteolytic treatment (row), mean values followed by the common letter do not differ significantly (*p* < 0.05). Mean ± standard deviation. *** (*p* < 0.001).

**Table 5 molecules-25-04692-t005:** Results of reducing power (*A*_700_).

Sample	Nondigested Extract	Pepsin Hydrolysate	Pepsin/Pancreatin Hydrolysate	Sample (S)	Treatment (T)	S × T
S1	0.317 ± 0.053 ^Aa^	0.832 ± 0.048 ^Ba^	0.333 ± 0.057 ^Aa^	***	***	***
S2	0.630 ± 0.042 ^Ab^	0.824 ± 0.053 ^Ba^	0.414 ± 0.071 ^Cb^			
S3	0.343 ± 0.048 ^Aac^	0.723 ± 0.096 ^Bab^	0.338 ± 0.047 ^Aa^			
S4	0.579 ± 0.087 ^Abd^	1.084 ± 0.174 ^Bc^	0.417 ± 0.055 ^Aab^			
S5	0.420 ± 0.038 ^Ac^	0.904 ± 0.058 ^Bad^	0.201 ± 0.017 ^Cc^			
S6	0.551 ± 0.021 ^Ad^	1.022 ± 0.067 ^Bcd^	0.233 ± 0.029 ^Cc^			
SC	0.572 ± 0.033 ^Abd^	0.648 ± 0.063 ^Bb^	0.330 ± 0.035 ^Ca^			

^a–d^ Within the same batches (column), mean values followed by the common letter do not differ significantly (*p* < 0.05). ^A–C^ Within the proteolytic treatment (row), mean values followed by the common letter do not differ significantly (*p* < 0.05). Mean± standard deviation. *** (*p* < 0.001).

**Table 6 molecules-25-04692-t006:** Results of angiotensin-converting enzyme (ACE-I) inhibiting activity (%).

Sample	Nondigested Extract	Pepsin Hydrolysate	Pepsin/Pancreatin Hydrolysate	Sample (S)	Treatment (T)	S × T
S1	40.54 ± 1.85 ^Aa^	42.41 ± 1.92 ^Aa^	43.69 ± 1.72 ^Aa^	***	***	***
S2	39.21± 1.10 ^Aa^	51.70 ± 2.26 ^Bb^	44.11 ± 0.90 ^Ca^			
S3	42.39 ± 1.68 ^Aa^	43.92 ± 2.66 ^Aa^	40.37 ± 0.52 ^Ab^			
S4	44.61 ± 1.34 ^Abc^	54.32 ± 2.46 ^Bb^	47.97 ± 1.04 ^Ac^			
S5	43.72± 1.92 ^Aac^	52.17 ± 1.92 ^Bb^	43.57 ± 1.19 ^Aa^			
S6	43.33 ± 0.24 ^Aab^	52.09 ± 0.74 ^Bb^	49.51 ± 0.95 ^Cc^			
SC	47.28 ± 1.86 ^Ac^	61.17 ± 1.66 ^Bc^	57.32 ± 1.46 ^Cd^			

^a–d^ Within the same batches (column), mean values followed by the common letter do not differ significantly (*p* < 0.05). ^A–C^ Within the proteolytic treatment (row), mean values followed by the common letter do not differ significantly (*p* < 0.05). Mean ± standard deviation. *** (*p* < 0.001).

**Table 7 molecules-25-04692-t007:** Details of the experimental tests.

Sample	Treatment
S1	marinated in sour milk
S2	marinated in sour milk + addition of herbs
S3	ultrasonic level I + marinated in sour milk
S4	ultrasonic level I + marinated in sour milk + addition of herbs
S5	ultrasonic level II + marinated in sour milk
S6	ultrasonic level II + marinated in sour milk + addition of herbs
SC	control sample (raw meat)

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
