# Peer review of "Influence of Sonication and Taraxacum Officinale Addition on the Antioxidant and Anti-ACE Activity of Protein Extracts from Sous Vide Beef Marinated with Sour Milk and after In Vitro Digestion"

_molecules, 2020, doi:10.3390/molecules25204692_

Round 1

Reviewer 1 Report

Dear Author, I reviewed the manuscript  (molecules-929193) entitled: Influence of sonication and Taraxacum officinale addition on the antioxidant and anti-ACE activity of protein extracts from sous-vide beef marinated with sour milk and after in vitro digestion. This manuscript presents relevant information about the antioxidant potential of protein animal extracts. However, some sections of the discussion of the obtained findings can be improved. For this reason, I considered that this manuscript needs minor changes for being considered for its publication in this journal. 

Additional comments:
Try to update some references in this manuscript. 

Highlight the importance of study these bioactive properties of these proteins extracts. 

Check weight units in this manuscript to use the same form.

Check the extension of long paragraphs in this manuscript. 

Try to include future trends to keep working with the obtained findings.

Try to conclude with a general statement of the most relevant findings of this research.  

Author Response

Reviewer 1

Dear Author, I reviewed the manuscript  (molecules-929193) entitled: Influence of sonication and Taraxacum officinale addition on the antioxidant and anti-ACE activity of protein extracts from sous-vide beef marinated with sour milk and after in vitro digestion. This manuscript presents relevant information about the antioxidant potential of protein animal extracts. However, some sections of the discussion of the obtained findings can be improved. For this reason, I considered that this manuscript needs minor changes for being considered for its publication in this journal. 

Additional comments:

Suggestion 1.Try to update some references in this manuscript. 

Thank you for this suggestion. Indeed, an additional literature review was necessary and provided support to highlight our results. The manuscript was enriched with the following references:

Mehta, N., Ahlawat, S. S., Sharma, D. P., & Dabur, R. S. (2015). Novel trends in development of dietary fiber rich meat products—a critical review. Journal of Food Science and Technology, 52(2), 633-647.

Domínguez, R., Pateiro, M., Gagaoua, M., Barba, F. J., Zhang, W., & Lorenzo, J. M. (2019). A comprehensive review on lipid oxidation in meat and meat products. Antioxidants, 8(10), 429.

Alarcon-Rojo, A. D., Carrillo-Lopez, L. M., Reyes-Villagrana, R., Huerta-Jiménez, M., & Garcia-Galicia, I. A. (2019). Ultrasound and meat quality: A review. Ultrasonics sonochemistry, 55, 369-382.

Boateng, E. F., & Nasiru, M. M. (2019). Applications of ultrasound in meat processing technology: a review. Food Science and Technology, 7(2), 11-15.

Vidal, A. R., Cechin, C. D. F., Cansian, R. L., Mello, R. D. O., Schmidt, M. M., Demiate, I. M., ... & Dornelles, R. C. P. (2018). Enzymatic hydrolysis (pepsin) assisted by ultrasound in the functional Properties of hydrolyzates from different collagens. Ciência Rural, 48(3).

Vidal, A. R., Cansian, R. L., Mello, R. D. O., KUBOTA, E. H., Demiate, I. M., Zielinski, A. A. F., & Dornelles, R. C. P. (2020). Effect of ultrasound on the functional and structural properties of hydrolysates of different bovine collagens. Food Science and Technology, 40(2), 346-353.

Ryan J. T., Ross R. P., Bolton D., Fitzgerald G. F., Stanton C. (2011). Bioactive peptides from muscle sources: meat and fish. Nutrients, 3, 765-791

Bhat Z. F., Kumar S., Bhat H. F. (2015). Bioactive peptides of animal origin: a review. Journal of Food Science and Technology, 52, 5377-5392

Suggestion 2. Highlight the importance of study these bioactive properties of these proteins extracts. 

Thank you for this tip. Indeed, we now see the need to emphasize the importance of bioactive peptides from meat, especially in the context of health protection. Therefore we supplemented our manuscript with the following text entry: “Contemporary knowledge in the field of meat technology indicates a number of possible directions of modification of the nutritional and health-promoting value of meat products. This can be achieved, for example, through innovative technological treatments (eg sonication), adding an ingredient with high pro-health potential (eg dandelion herb) or discovering new properties of food ingredients, such as biologically active peptides. In this context, these peptides can be used as new ingredients in the development of functional food products. Food-derived bioactive peptides contain a sequence of 2-20 amino acids and can exhibit various in vivo effects. Among the many types of biopeptide activity, their antioxidant and angiotensin converting enzyme (ACE) inhibitory properties are the most frequently studied due to their potential in preventing chronic non-communicable diseases. Peptides acting as antioxidants may contribute to the maintenance of the oxidative stability of the meat tissue. They are also involved in preventing the negative effects of oxidative stress. This is due to the excessive accumulation of reactive oxygen species in the body and can lead to pathological conditions manifested by cardiovascular diseases, diabetes and other metabolic disorders. In turn, peptides with ACE inhibitor activity contribute to the reduction of blood pressure in patients with arterial hypertension, which is a risk factor for the development of cardiovascular diseases.The study assessed the effect of peptides obtained from sous-vide beef steaks in relation to the above-mentioned bioactivities. It was found that the proteolytic changes caused by the applied technological measures favored the release of peptides acting as antioxidants and inhibiting the action of ACE.”

We would like to thank you again for this suggestion. Thanks to the introduced changes, the scientific value of our manuscript has increased.

Suggestion 3. Check weight units in this manuscript to use the same form.

Thank you for pointing out our mistake. I would like to clarify that the inconsistent record of the units resulted from our oversight. Therefore, we have re-edited the text and all units have been unified directly in the text. We would like to thank you again for your thorough review of the presented manuscript.

Suggestion 4. Check the extension of long paragraphs in this manuscript. 

As suggested by the reviewer, the text was re-edited and long paragraphs were shortened. We made all the changes directly in the text. It was a good change. The revised version of the manuscript is more understandable and accessible to the reader.

Suggestion 5 and 6. Try to include future trends to keep working with the obtained findings. Try to conclude with a general statement of the most relevant findings of this research.  

Thank you for this tips. The authors changed the conclusion taking into account the reviewer's suggestions: “The results of the present study showed that there is a possibility to produced potentially functional meat products as a result of the use of ultrasound in combination with dandelion extracts. The potentially functional characteristics of meat product probably derive from the presence of biologically active peptides increased by sonication and polyphenols from T. officinale. The use of a T. officinale additive during sous-vide processing enhances the antioxidant potential of compounds present in beef, such as peptides and amino acids. Additionally, the use of dandelion increases the potential of beef steaks prepared with the sous-vide method against oxidative factors. The use of ultrasound, affects the intensity of proteolytic changes of the product. It was observed that the interaction of ultrasound treatment during the maceration of meat with sour milk may contribute to an increase in the antioxidant potential and inhibition of ACE. Importantly, the obtained protein extracts also show biological activity under the influence of proteolytic enzymes, which are the equivalent of human digestive enzymes. Therefore, the results of the present study may have future applications for the production of healthier meat-based products.”

Reviewer 2 Report

Influence of sonication and Taraxacum officinale addition on the antioxidant and anti-ACE activity of protein extracts from sous-vide beef marinated with sour milk and after in vitro digestion

The topic of the manuscript is very interesting for the journal and contributes to the
knowledge of natural antioxidants as well as emerging processing technologies.

The work has good scientific quality due to the analysis carried out, however, English should be revised to allow a better understanding of the work. For this reason and together with the clarifications requested below, I recommend minor revision.

- LINE 25. The inclusion of more words would allow to better represent the work.

- LINE 28. Change “want” by “demand”.

- LINE 37. Include a reference to justify this statement.

- LINE 56. Change “The interest in the use of ultrasound in meat processing technology results from…” by “The interest in the use of ultrasound technology in meat processing results from…”

- LINES 260-341. Divide discussion into sections that facilitate understanding. Although the discussion with results made with samples of shrimp and cheese are interesting, references more related to the study matrix should be used.

- LINE 264. I'm confused. Was the work done with beef or pork?

- LINE 364. What is the meaning of (m/m)?

- LINE 414. Add the brand and model of the spectrophotometer used.

- TABLE 1. Check the table. There is an error in the value of EAA/NEAA for S2.

- TABLE 6. Check the table. Standard deviation (SD) values of pepsin hydrolysate are missing.

Author Response

Reviewer 2

Influence of sonication and Taraxacum officinale addition on the antioxidant and anti-ACE activity of protein extracts from sous-vide beef marinated with sour milk and after in vitro digestion

The topic of the manuscript is very interesting for the journal and contributes to the
knowledge of natural antioxidants as well as emerging processing technologies.

The work has good scientific quality due to the analysis carried out, however, English should be revised to allow a better understanding of the work. For this reason and together with the clarifications requested below, I recommend minor revision.

Suggestion 1: LINE 25. The inclusion of more words would allow to better represent the work.

Thank you for this opinion. Indeed, placing additional keywords is a valid suggestion. This will allow interested persons to find the results of our research better. By expanding this section, our work will be able to reach other researchers faster. We decided to include additional descriptions in the text, ie beef, antioxidant and ACE inhibitor. Thanks again for your valuable suggestion

Suggestion 2: LINE 28. Change “want” by “demand”.

Thank you for this suggestion, the changes have been made directly in the text.

Suggestion 3: LINE 37. Include a reference to justify this statement.

Thank you for this tip. In accordance with the opinion of the reviewer, the references have been supplemented with the following items:

Mehta, N., Ahlawat, S. S., Sharma, D. P., & Dabur, R. S. (2015). Novel trends in development of dietary fiber rich meat products—a critical review. Journal of Food Science and Technology, 52(2), 633-647.

Suggestion 4: LINE 56. Change “The interest in the use of ultrasound in meat processing technology results from…” by “The interest in the use of ultrasound technology in meat processing results from…”

Thank you for this suggestion, the changes have been made directly in the text.

Suggestion 5: LINES 260-341. Divide discussion into sections that facilitate understanding. Although the discussion with results made with samples of shrimp and cheese are interesting, references more related to the study matrix should be used.

Thank you for this opinion. The cited examples show the trend observed during food sonication. In view of this, it was decided to keep this entry for milk products (Cheddar cheese) as well as shrimp. Nevertheless, we cannot disagree with the Reviewer's opinion that the text should be enriched with other examples concerning the tested matrix (bovine/pork collagen). Therefore, we reviewed the literature again and supplemented the discussion with new examples.

Suggestion 6:  LINE 264. I'm confused. Was the work done with beef or pork?

Thank you for pointing out our mistake. Indeed, there is an incorrect description in the text. This is due to our oversight. Of course, the authors meant beef (not pork). We wanted to assure you that we did not want to mislead our readers. Thanks again for such a thorough review of our manuscript.

Suggestion 7: LINE 364. What is the meaning of (m/m)?

Thank you for this suggestion. An error has occurred due to language barriers. Of course, it was about the unit w/w (not m/m). The correct spelling was placed directly in the text.

Suggestion 8:  LINE 414. Add the brand and model of the spectrophotometer used.

Thank you for this suggestion. It was U-5100 UV-Vis, Hitachi HighTechnologies America Inc., Schaumburg, IL, USA.  The changes have been made directly in the text.

Suggestion 9: TABLE 1. Check the table. There is an error in the value of EAA/NEAA for S2.

Thank you for this valuable suggestion. We did indeed make a mistake. It was a value of 0.456 (not 0.0456). The notation was corrected directly in the manuscript. We are sorry that we did not notice our mistake earlier. Therefore, thank you again for pointing out our mistake.

Suggestion 10. TABLE 6. Check the table. Standard deviation (SD) values of pepsin hydrolysate are missing.

Thanks again for the suggestion. There was an error in the editing of the text. In the new version of the manuscript, the missing data has been completed.

Reviewer 3 Report

On the manuscript " Influence of sonication and Taraxacum officinale  addition on the antioxidant and anti-ACE activity of  protein extracts from sous-vide beef marinated with

sour milk and after in vitro digestion"  by Paulina Keska et al. the authors assessed the effect of pretreating beef as a raw material for sous-vide steak preparation. The pretreatment involved maceration of a batch of meat in sour milk with the simultaneous use of ultrasound (250 or 500 W) as well as the addition of Taraxacum officinale. The biological activity profile of the peptides was assessed in terms of their antioxidant activity and the inhibiting activity against angiotensin-converting enzyme (ACE).

The authors decide plant raw materials in the production of beef steaks prepared by sous-vide methods. It is assumed that the procedure used will enhances the antioxidant potential of compounds present in beef, such as peptides and amino acids. The results showed that additional use of technological treatments, namely marinating in sour milk or the use of ultrasound, potentially affects the intensity of biochemical changes (including proteolytic changes of the raw  material), thus affecting the degradation of proteins and modifying their biological activity against oxidation (based on ABTS, Fe2+ chelation, and reducing power (RP) tests) factors and angiotensin converting enzyme (ACE) inhibiting properties.

SOME COMMENTS

The paper fit the aims and scope of the Molecules. The title is informative and give a clear idea of the paper.

The work is technically sound and scientifically valid. The used methodology is appropriate and the objectives of the work in addition to the experimental design are well planned and followed. The conclusions drawn are fully supported by the data presented and the claims are appropriately discussed in the context of previous literature. In addition, the paper provided sufficient methodological detail that the experiments can be reproduced.

I would like to considerate for publication after the minor modifications were made.

ABSTRACT

Abstract is adjusted to the developed work, to the used methodologies and to the obtained results.

INTRODUCTION

The introduction is interesting, well designed and structured, and adjusted to the target subject .

The transition between Line 131 –132 seems confusing. Please correct the sentences.

On Table 1 the amino acids abbreviations are missing, besides be universal. All the abbreviations must be defined (EAA/NEAA, S1 to S6,)

Caption of table 2: Change “ … Protein digestibility [%] from … “ by “ .. Protein digestibility (%) from … “

The same correction for Table 4 Cation “Results of Fe2+ 205 chelating activity (%)”.

On Tables 1 - 4 the letters representing the significant values must be all lower case letters. The authors present lower case and upper case letters. Correct!

On Footnotes of Table 2 change “ … *** (p<0.001), ** (p><0.01), * (p><0.001); ** (p<0.01); * (p<0.001), ** (p><0.01), * (p><0.05).” by “ … *** (p<0.001), ** (p><0.01), * (p><0.001); * (p<0.001), ** (p><0.01), * (p><0.05).” since ** is not present in the table.

On Foot notes of Tables 3 6to Table 6 delete ** (p<0.01) and * (p<0.001), ** (p><0.01), * (p><0.05), because on the Tables only *** is considered.

MATERIALS AND METHODS

This section is adjusted.

RESULTS AND DISCUSSION

This section is well structured and discussed, highlighting the most important results and findings.

Conclusions

This section can be improved highlighting the most important results

   In my opinion this paper will achieve the scientific quality standards needed to be published in Molecules after the proposed corrections.

Author Response

Reviewer 3

On the manuscript " Influence of sonication and Taraxacum officinale  addition on the antioxidant and anti-ACE activity of  protein extracts from sous-vide beef marinated with

sour milk and after in vitro digestion"  by Paulina Keska et al. the authors assessed the effect of pretreating beef as a raw material for sous-vide steak preparation. The pretreatment involved maceration of a batch of meat in sour milk with the simultaneous use of ultrasound (250 or 500 W) as well as the addition of Taraxacum officinale. The biological activity profile of the peptides was assessed in terms of their antioxidant activity and the inhibiting activity against angiotensin-converting enzyme (ACE).

The authors decide plant raw materials in the production of beef steaks prepared by sous-vide methods. It is assumed that the procedure used will enhances the antioxidant potential of compounds present in beef, such as peptides and amino acids. The results showed that additional use of technological treatments, namely marinating in sour milk or the use of ultrasound, potentially affects the intensity of biochemical changes (including proteolytic changes of the raw  material), thus affecting the degradation of proteins and modifying their biological activity against oxidation (based on ABTS, Fe2+ chelation, and reducing power (RP) tests) factors and angiotensin converting enzyme (ACE) inhibiting properties.

 SOME COMMENTS

The paper fit the aims and scope of the Molecules. The title is informative and give a clear idea of the paper.

The work is technically sound and scientifically valid. The used methodology is appropriate and the objectives of the work in addition to the experimental design are well planned and followed. The conclusions drawn are fully supported by the data presented and the claims are appropriately discussed in the context of previous literature. In addition, the paper provided sufficient methodological detail that the experiments can be reproduced.

I would like to considerate for publication after the minor modifications were made.

 Suggestion 1: The transition between Line 131 –132 seems confusing. Please correct the sentences.

Thank you for this opinion. Indeed, after reconsidering, the fragment  (line 131-132) included in the work may be incomprehensible. That is why we decided to edit the text. Some of the important information was included in the text (two-way ANOVA results), and these results were removed from Fig. 1. Thanks to this procedure, the presented data is more readable and understandable for a potential reader. Therefore, we agree with the reviewer's suggestion that such changes was necessary.

Suggestion 2: On Table 1 the amino acids abbreviations are missing, besides be universal. All the abbreviations must be defined (EAA/NEAA, S1 to S6,).

As suggested by the reviewer, Table 1 was supplemented with the list of abbreviations used, i.e. the list of determined amino acids and the system of samples (S1-S6). The abbreviation EAA (Essentials amino acids) and NEAA (Nonessential amino acids) is explained in the headings of table 1, therefore we decided not to explain these abbreviations again in the list below the table.

Suggestion 3: Caption of table 2: Change “ … Protein digestibility [%] from … “ by “ .. Protein digestibility (%) from … “

Thank you for this suggestion, the changes have been made directly in the text

Suggestion 4: The same correction for Table 4 Cation “Results of Fe2+ 205 chelating activity (%)”.

Again, thank you for this tip. All changes have been made directly in the text

Suggestion 5: On Tables 1 - 4 the letters representing the significant values must be all lower case letters. The authors present lower case and upper case letters. Correct!

Thank you for pointing out our mistake. It results from our oversight. indeed, the coding of the similarities between the trials was incorrectly marked. Of course the lowercase letters refer to the similarities between the columns and the uppercase letters between the rows. In the new version of the manuscript, we corrected our error in Tables 2-6. We would like to thank the Reviewer again for his thorough review of our manuscript.

Suggestion 6: On Footnotes of Table 2 change “ … *** (p<0.001), ** (p><0.01), * (p><0.001); ** (p<0.01); * (p<0.001), ** (p><0.01), * (p><0.05).” by “ … *** (p<0.001), ** (p><0.01), * (p><0.001); * (p<0.001), ** (p><0.01), * (p><0.05).” since ** is not present in the table.

On Foot notes of Tables 3 6to Table 6 delete ** (p<0.01) and * (p<0.001), ** (p><0.01), * (p><0.05), because on the Tables only *** is considered.

Thank you for this tip. As suggested by the Reviewer, unnecessary data has been removed from the footnotes. This makes the text clearer. We believe that in the current version it will be easier to read for those interested. Therefore, we would like to thank you again for this opinion.

 Suggestion 7. Conclusions. This section can be improved highlighting the most important results

  Thank you for this tips. The authors changed the conclusion taking into account the reviewer's suggestion: “The results of the present study showed that there is a possibility to produced potentially functional meat products as a result of the use of ultrasound in combination with dandelion extracts. The potentially functional characteristics of meat product probably derive from the presence of biologically active peptides increased by sonication and polyphenols from T. officinale. The use of a T. officinale additive during sous-vide processing enhances the antioxidant potential of compounds present in beef, such as peptides and amino acids. Additionally, the use of dandelion increases the potential of beef steaks prepared with the sous-vide method against oxidative factors. The use of ultrasound, affects the intensity of proteolytic changes of the product. It was observed that the interaction of ultrasound treatment during the maceration of meat with sour milk may contribute to an increase in the antioxidant potential and inhibition of ACE. Importantly, the obtained protein extracts also show biological activity under the influence of proteolytic enzymes, which are the equivalent of human digestive enzymes. Therefore, the results of the present study may have future applications for the production of healthier meat-based products.”

Round 2

Reviewer 3 Report

On the manuscript " Influence of sonication and Taraxacum officinale addition on the antioxidant and anti-ACE activity of protein extracts from sous-vide beef marinated with

sour milk and after in vitro digestion" by Paulina Keska et al. the authors assessed the effect of pretreating beef as a raw material for sous-vide steak preparation. The pretreatment involved maceration of a batch of meat in sour milk with the simultaneous use of ultrasound (250 or 500 W) as well as the addition of Taraxacum officinale. The biological activity profile of the peptides was assessed in terms of their antioxidant activity and the inhibiting activity against angiotensin-converting enzyme (ACE). The authors decide plant raw materials in the production of beef steaks prepared by sous-vide methods. It is assumed that the procedure used will enhances the antioxidant potential of compounds present in beef, such as peptides and amino acids. The results showed that additional use of technological treatments, namely marinating in sour milk or the use of ultrasound, potentially affects the intensity of biochemical changes (including proteolytic changes of the raw material), thus affecting the degradation of proteins and modifying their biological activity against oxidation (based on ABTS, Fe2+ chelation, and reducing power (RP) tests) factors and angiotensin converting enzyme (ACE) inhibiting properties.

SOME COMMENTS

The paper fit the aims and scope of the Molecules. The title is informative and give a clear idea of the paper.

The work is technically sound and scientifically valid. The used methodology is appropriate and the objectives of the work in addition to the experimental design are well planned and followed. The conclusions drawn are fully supported by the data presented and the claims are appropriately discussed in the context of previous literature. In addition, the paper provided sufficient methodological detail that the experiments can be reproduced.

In the revised version of the manuscript the authors done the corrections / alterations suggested by the reviewer improving significantly the manuscript, therefore in my opinion this version of the manuscript achieved the scientific standard to be published im Molecules journal from MDPI.

Author Response

Thank you very much for accepting the changes and additions introduced
in the publication.